# Reproducibility Report: Deep Fair Clustering for Visual Learning

## 1 Reproducibility Summary

### 2 Scope of Reproducibility

3 Deep Fair Clustering (DFC) aims to provide a clustering algorithm that is fair, clustering-favourable, and which can be
4 used on high-dimensional and large-scale data. In existing frameworks there is a trade-off between clustering quality
5 and fairness. In this report we aim to reproduce a selection of the results of DFC; using two of four datasets and all four
6 metrics that were used in the original paper, namely accuracy, Normalized Mutual Information (NMI), balance and
7 entropy. We use the authors' implementation and check whether it is consistent with the description in the paper. As
8 extensions to the original paper we look into the effects of 1) using no pretrained cluster centers, 2) using different
9 divergence functions as clustering regularizers and 3) using non-binary/corrupted sensitive attributes.

### 10 Methodology

11 The open source code of the authors has been used. The datasets and data-preprocessing has been done with our code,
12 since the authors did not provide the datasets in their code. Also the pretrained Variational Autoencoder (VAE) dataset
13 had to be re-implemented for the *Color Reverse MNIST* . For the extensions we wrote extra functions. For measuring
14 the influence of discarding the pretrained cluster centers, the code was already provided by the authors.

### 15 Results

16 For the *MNIST-USPS* dataset, we report similar accuracy and NMI values that are within 1.2% and 0.5% of the values
17 reported in the original paper. However, the balance and entropy differed significantly, where our results were within
18 73.1% and 30.3% of the original values respectively. For the *Color Reverse MNIST* dataset, we report similar values on
19 accuracy, balance and entropy, which are within 5.3%, 2.6% and 0.2% respectively. Only the value of the NMI differed
20 significantly, name within 12.9% of the original value In general, our results still support the main claim of the original
21 paper, even though on some metrics the results differ significantly.

### 22 What was easy

23 The open source code of the authors was beneficial; it was well structured and ordered into multiple files. Furthermore,
24 the code to use randomly initialized instead of pretrained cluster centers was already provided.

### 25 What was difficult

26 First of all, the main difficulty in reproducing the paper was caused by the coding style; due to the lack of comments
27 it was difficult to get a good understanding of the code. Secondly, we were required to download the data ourselves.
28 However, these filenames and labels did not correspond to the included txt-files by the authors. Therefore, the model did
29 not learn and we regenerated `train_mnist.txt` and `train_usps.txt`. Finally, the authors only included pretrained
30 models for the *MNIST-USPS* dataset. As a consequence, we had to pre-train some parts of the DFC algorithm for the
31 *Color Reverse MNIST* dataset.

# 1 Introduction

With the increased application of Machine Learning in automated systems, particularly in decision making systems, it has become desirable that individuals are treated equally in such automated environments. However, there exists a trade-off between the fairness and the performance of machine learning algorithms in a given task [Li et al., 2020]. In current fair clustering algorithms, fair and effective representations are learned by mainly using small-scale and low-dimensional data.

Deep Fair Clustering (DFC) is an algorithm that aims to learn fair and clustering-favorable representations for large-scale and high-dimensional data. In this context, feature representations are considered to be fair if they are statistically independent of sensitive attributes. DFC consists of an encoder that produces the representations, and a discriminator that tries to predict the value of the sensitive attribute of a representation. A minimax game is used to learn fair representations in an adversarial manner. In order to preserve the utility of the representations, clustering is performed on all datapoints with the same sensitive attribute. This component is called 'structural preservation' because it preserves the clustering structure in each sensitive attribute. Finally, The KL-divergence is used as a clustering regularizer to prevent the formation of large clusters.

All code is available on Github [Al Gerges et al.].

# 2 Scope of reproducibility

The goal of this work is to validate the reproducibility of the DFC algorithm proposed by Li et al. [2020] beyond the scope of the original paper. The main claims of the original paper are as follows:

> Claim 1: **DFC produces a fair clustering partition on high dimensional and large-scale visual data.**

> Claim 2: **DFC produces clustering-favorable representations under a fairness constraint.**

To test the validity of claim 1, the balance and entropy scores will be examined and compared with the original paper. The validity of claim 2 will be tested similarly, where we instead examine the accuracy and normalized mutual information (NMI) score. Important to note is that the original paper mainly evaluated the DFC algorithm on binary sensitive attributes. As an example, in Li et al. [2020] a sensitive attribute was defined as whether an image from the *MNIST* dataset has been reversed or not. Generally speaking, in the original paper the sensitive attributes could only take one out of two possible values. However, sensitive attributes in the real world, like race or gender, can take on multiple variables.

Thus, to evaluate the robustness of claim 1, we will perform the DFC algorithm for non-binary sensitive attributes. These will be modelled by 'corrupting' the images in the *Color Reverse MNIST* dataset. The corruption process is done by partially replacing the pixels in the background of the images. For the *MNIST* dataset, some pixels will be whitened, while for the *Color Reverse MNIST* dataset the some pixels will be replaced by black pixels.

Furthermore, we will investigate the robustness of both claims by testing the DFC algorithm on different model configurations. Specifically, we will test out different clustering regularizers by replacing the KL-divergence with other divergence measures, namely the Jensen-Shannon divergence (JS-divergence) and the Cauchy-Schwarz divergence (CS-divergence).

Finally, in the original paper it is mentioned that pretrained cluster centers were used in the DFC algorithm. However, the motivation of using pretrained cluster centers in DFC is omitted, which might suggest that pre-training cluster centers are not a necessary part of the DFC pipeline. Therefore, we will examine the influence of pretrained cluster centers in DFC.

# 3 Methodology

## 3.1 Model descriptions

Li et al. [2020] use a pretrained convolutional variational autoencoder (VAE). The available code only contained the pretrained encoder and decoder for the *MNIST-USPS* dataset [Li et al.]. We implemented and pretrained a convolutional VAE for the *Color Reverse MNIST* dataset. The encoder is build of four convolutional layers, followed by batch normalization and a ReLU activation function. Moreover, the decoder is implemented by reversing the layers of the

78 encoder. Both the encoder and decoder contained 610K and 58.9K parameters respectively. The VAE is trained using
79 the Adam-optimizer and a learning rate of $1e-3$.

80 Li et al. [2020] also used pretrained cluster centers to start their DFC algorithm off with high accuracy clusters. They
81 only provided pretrained cluster centers for the *MNIST-USPS* dataset: Therefore, in order to reproduce the results, we
82 were required to obtain pretrained cluster centers for the *Color Reverse MNIST* dataset. For this task we used k-means
83 clustering[1] with $k = 10$. Because the original code of the authors used 64-dimensional cluster centers, we first scaled
84 our $32 \times 32$ images down with a max pooling layer with 4 sized filters, so that the images would go from $32 \times 32$ to
85 $8 \times 8$. After dimension reduction every image becomes a $1 \times 64$ vector. We then fit every image in the dataset using
86 MiniBatchKMeans from the sklearn package[2]. With `max_iter` $= 1000$ and `batch_size` $= 512$. This results in our
87 pretrained cluster centers which can be trained for every dataset.

88 To examine during clustering whether fair representations are reached, a discriminator is used; when it cannot distinguish
89 based on the sensitive attribute the representations are fair. This discriminator is a multilayer perceptron (MLP) using
90 three linear layers, of which the first two are followed by a ReLU activation function and a dropout of 0.5: the final
91 layer is followed by a sigmoid activation function. The discriminator is trained jointly with the encoder for 20000
92 epochs. Finally, the Adam optimizer is used with an initial learning rate of $lr_{\text{init}} = 1e-4$. The learning rate is adjusted
93 with $lr = lr_{\text{init}}(1 + 10t)^{-0.75}$, with $t = 0$ at the start of the training process; with every iteration $t$ is linearly increased
94 to $t = 1$ at the end of the training process.

95 The objective function consists of three parts; the fairness-adversarial loss ($\mathcal{L}_f$), the structural preservation loss ($\mathcal{L}_s$)
96 and the clustering regularizer term ($\mathcal{L}_c$). The task of the fairness-adversarial loss is to minimize the divergence between
97 the cluster assignments of the different subgroups. In this way the term promotes a similar cluster distribution for all
98 subgroups, hence, statistical independence between cluster assignments and the particular protected subgroup that the
99 sample belongs to. The fairness-adversarial loss can be written as:

$$\mathcal{L}_f := \mathcal{L}(\mathcal{D} \circ \mathcal{A} \circ \mathcal{F}(X), G), \tag{1}$$

100 where $\mathcal{L}$ denotes the cross-entropy loss and $\circ$ denotes the function composition: moreover, $\mathcal{D}$, $\mathcal{A}$, $\mathcal{F}$ denotes the
101 discriminator, cluster assignment and encoder respectively.

102 The fairness-adversarial loss encourages statistical independence of the cluster assignments and the sensitive attribute
103 G, however, only optimizing $\mathcal{L}_f$ is not enough as it can lead to a degenerate solution, where the representations that
104 are produced by the encoder are all constant. Of course, such a constant representation cannot lead to good clustering
105 quality; it would hide, rather than illuminate, the fundamental structure in the data. The structural preservation loss
106 prevents such a solution by penalizing it when the inner structure of a particular subgroup is altered in the DFC setting,
107 as opposed to clustering the subgroup individually. The preservation loss, which was proposed by the authors [Li et al.,
108 2020] is given as follows:

$$\mathcal{L}_s := \sum_{g \in [M]} \left\| \hat{P}_g \hat{P}_g^T - P_g P_g^T \right\|^2, \tag{2}$$

109 where $[M]$ denotes the set of sensitive attributes, $\hat{P}_g$ and $P_g$ denote the (soft) assignments of the $g-$th protected
110 subgroup when individually clustered and clustered with DFC, respectively.

111 Following other work in deep clustering, DFC employs a clustering regularizer to strengthen prediction confidence
112 and to prevent large cluster sizes [Li et al., 2020]. Contrary to earlier work, the clustering regularizer is chosen in such
113 away that it encourages the members of a particular protected subgroup to be distributed equally over the clusters. To
114 increase the confidence of the prediction an auxiliary target distribution $Q$ is defined. This target distribution is defined
115 in such a way that it favors current high confidence assignments and is calculated as:

$$q_k = \frac{(p_k)^2 / \sum_{x \in X_g} p_k}{\sum_{k' \in [K]} ((p_{k'})^2 / \sum_{x \in X_g} p_{k'})}, \tag{3}$$

116 with $p_k$ the probability that sample $x$ belongs to cluster $k$, and $X_g$ the samples that belong to protected subgroup $G$.
117 Then, the clustering regularizer loss is defined as the KL-divergence between soft assignment $P$ and auxiliary target
118 distribution $Q$:

$$\mathcal{L}_c := KL(P||Q) = \sum_{g \in [M]} \sum_{x \in X_g} \sum_{k \in [K]} p_k \log \frac{p_k}{q_k}. \tag{4}$$

---

[1] https://scikit-learn.org/stable/modules/generated/sklearn.cluster.KMeans.html
[2] https://scikit-learn.org/stable/modules/generated/sklearn.cluster.MiniBatchKMeans.html

Again following the literature, the authors have chosen to use the Student t-distribution for soft cluster assignment [Li et al., 2020]. The probability that the representation $z$ (corresponding to a particular sample $x$) belongs to cluster $c_k$ is then given by:

$$p_k = \frac{(1 + \frac{1}{\alpha}||z - c_k||^2)^{-\frac{\alpha+1}{2}}}{\sum_{k' \in [K]}(1 + \frac{1}{\alpha}||z - c_{k'}||^2)^{-\frac{\alpha+1}{2}}}, \tag{5}$$

with $\alpha$ the degree of freedom of the Student's t-distribution. In conclusion, the overall objective is defined as the following minimax strategy:

$$\max_{\mathcal{F},\mathcal{A}} \quad \alpha_f \mathcal{L}_f - \alpha_s \mathcal{L}_s - \mathcal{L}_c, \tag{6}$$

$$\min_{\mathcal{D}} \quad \alpha_f \mathcal{L}_f \tag{7}$$

with $\alpha_f$ and $\alpha_s$ as trade-off hyperparameters.

## 3.2 Datasets

In this study, we have used two publicly available datasets: *Color Reverse MNIST* and *MNIST-USPS* datasets. Both datasets contain a collection of grey-scale images of hand-written digits (0-9).

The first dataset, *MNIST-USPS* , is a combination of the *MNIST*[3] and *USPS*[4] dataset. Both, *MNIST* and *USPS* are downloaded using the `torch.vision.dataset` package. The label distributions and total number of examples in the training and test set can be found in Table 1. The *MNIST* dataset contains approximately eight times more images than *USPS* . In the *MNIST-USPS* dataset, the source, either *MNIST* or *USPS* , is chosen to be the sensitive attribute.

The second dataset, *Color Reverse MNIST* , was constructed by reversing the images in the *MNIST* dataset and concatenating them to the original. The color reversed images were constructed with *pixel* = 255 - *pixel*. The label distributions and total number of examples in the training and test set can also be found in Table 1. Equivalent to the *MNIST-USPS* dataset, the sensitive attribute is the source of the image; in this case either *MNIST* or *Color Reverse MNIST* .

The images in all datasets are padded to create images of the same size ($32 \times 32$); this implies a padding of 2 and 8 for the images of *MNIST* and *USPS* respectively.

| Dataset | 0 | 1 | 2 | 3 | 4 | 5 | 6 | 7 | 8 | 9 | Total |
|---|---|---|---|---|---|---|---|---|---|---|---|
| *MNIST* train | 5923 | 6742 | 5958 | 6131 | 5842 | 5421 | 5918 | 6265 | 5851 | 5949 | 60000 |
| *USPS* train | 1194 | 1005 | 731 | 658 | 652 | 556 | 664 | 645 | 542 | 644 | 7291 |
| *Color Reverse MNIST* train | 11846 | 13484 | 11916 | 12262 | 11684 | 10842 | 11836 | 12530 | 11702 | 11898 | 120000 |
| *MNIST-USPS* train | 7117 | 7747 | 6689 | 6789 | 6494 | 5977 | 6582 | 6910 | 6393 | 6593 | 67291 |

Table 1: Label distribution per dataset

## 3.3 Extensions

### 3.3.1 Divergence Functions

As mentioned earlier in Section 2, we examined the effect of using different divergence functions as clustering regularizers by replacing the KL-divergence with either the Jensen-Shannon divergence (JS-divergence) or the Cauchy-Schwarz divergence (CS-divergence).

The JS-divergence is the smoothed and symmetric version of the KL-divergence and is calculated as follows:

$$JS(P||Q) = \frac{1}{2}KL(P||M) + \frac{1}{2}KL(Q||M) \tag{8}$$

where $M = \frac{1}{2}(P + Q)$ and $KL(.||.)$ is the KL-divergence as defined in 4.

---

[3]`http://yann.lecun.com/exdb/mnist/`
[4]`http://www.kaggle.com/bistaumanga/usps-dataset`

Furthermore, the CS-divergence is a divergence function that is inspired by information theory. It is given by the following (Jenssen et al. [2006]):

$$CS(P||Q) = -\log \frac{\int p(\mathbf{x})q(\mathbf{x})\mathrm{d}\mathbf{x}}{\sqrt{\int p^2(\mathbf{x})\mathrm{d}\mathbf{x} \int q^2(\mathbf{x})\mathrm{d}\mathbf{x}}} \tag{9}$$

The CS-divergence is, like the JS-divergence, a symmetric measure. Furthermore, the CS-divergence has the range $0 \leq CS(P||Q) \leq \infty$, where the minimum value of 0 is obtained if $p(\mathbf{x} = q(\mathbf{x}))$.

### 3.3.2 Corrupted Sensitive Attribute

Another extension mentioned in Section 2 is that we consider the influence of the corrupted sensitive attribute. In the *Color Reverse MNIST* dataset the presence of this attribute is clear in background color. Corrupting the sensitive attribute in this dataset implies random modifications in the background color. We compare two corruption rates (0.1 and 0.4) against the original images; for example, a rate of 0.1 implies that a random 10% of the background pixels are changed from black to white or vice versa.

### 3.3.3 Pretrained Cluster Centers

The final extension mentioned in Section 2 is that we would examine the influence of pretrained cluster centers on the performance of DFC. If no pretrained cluster centers were used, they would be randomly initialized with Xavier initialisation using a uniform distribution.

### 3.4 Evaluation

To evaluate the models, we used the four metrics that were also used by Li et al. [2020]: accuracy and Normalized Mutual Information (NMI) were used to evaluate the cluster validity, while balance and entropy were calculated to evaluate the fairness of DFC. Equations 10-13 are used to calculate the metrics: the NMI is calculated using sklearn.

$$Accuracy = \frac{\sum_{i=1}^{n} \mathbb{I}_{y_i = map(\hat{y}_i)}}{n} \tag{10}$$

$$NMI = \frac{\sum_{i,j} n_{ij} \log \frac{n \cdot n_{ij}}{n_{i+} \cdot n_{+j}}}{\sqrt{(\sum_i n_{i+} \log \frac{n_{i+}}{n})(\sum_j n_{+j} \log \frac{n_{+j}}{n})}} \tag{11}$$

$$Balance = \min_i \frac{\min_g |\mathcal{C}_i \cap X_g|}{n_{i+}} \tag{12}$$

$$Entropy = -\sum_i \frac{|\mathcal{C}_i \cap X_g|}{n_{i+}} \log \frac{|\mathcal{C}_i \cap X_g|}{n_{i+}} + \epsilon \tag{13}$$

In Eq. 10, $y_i$ and $\hat{y}_i$ represent the correct and predicted cluster label respectively: $map$ is a function that maps the cluster label $\hat{y}_i$ to the correct label $y_i$. In Eq. 11, $n_{ij}$ denotes the co-occurrence number; $n_{i+}$ and $n_{+j}$ denote the cluster size of the $i$-th and $j$-th clusters, in the obtained partition and ground truth, respectively. $n$ is the total data instance number. Furthermore, $\mathcal{C}_i$ represents the i-th cluster and $X_g$ the $g$-th protected subgroup. Finally, in Eq. 13, $\epsilon = 1e - 5$, to ensure the $\log$ will always be defined.

As mentioned before, accuracy and NMI are measures for the clustering quality. More specific, accuracy measures the correctness of clusters relative to a ground truth and NMI measures the similarity between the clustering obtained by DFC and the ground truth. For both metrics, a higher value indicates better clustering quality. Furthermore, balance and entropy evaluate the fairness of the obtained clustering. In particular, balance measures the homogeneity of the clustering across multiple sensitive attributes. A large value indicates that each cluster contains samples from multiple protected subgroups. If one cluster contains only instances of a particular protected subgroup, the balance has a score of 0. Entropy is a softer fairness metric than balance that measures the diversity of the clustering. Just like balance, a large entropy value indicates that samples from a protected subgroup are present in almost every cluster, which indicates a more fair clustering and thus more fair representations.

### 3.5 Computational requirements

The code was run locally on a GPU. The GPU in question is a GeForce GTX 970 with driver version 456.71. The CPU in this machine is an Intel Core i7-4770K. The memory used was 16.0 GB DDR3. For the main training of the adversarial network with 20000 iterations at 5000 iterations per evaluation the model ran in approximately 3.5 hours. This was the same computational cost to run DFC with a different divergence function. For the corruption extension we used 5000 iterations at 500 iterations per evaluation which took about 1.5 hours. The training of the VAE for the *Color Reverse MNIST* dataset took roughly 1 hour. The k-means clustering to obtain the pretrained clusters took approximately 15 minutes. Taking all this into account, the reproduction of the *Color Reverse MNIST* results from scratch took a total of circa 6.25 hours to compute. Finally, evaluating all the results with the saved models takes about 20 minutes. In conclusion, the code is not fast but it can be run on a local machine. A GPU is heavily recommended, because without one the code is about eight times slower.

## 4 Results

### 4.1 Reproduced Results

The original results from Li et al. [2020] as well as the reproduced results can be found in Table 2.

| Dataset | Method | Accuracy | NMI | Balance | Entropy |
|---------|--------|----------|-----|---------|---------|
| *Color Reverse MNIST* | Li et al. [2020] | 0.577 | 0.679 | 0.763 | 2.294/2.301 |
| | Reproduced | 0.548 | 0.591 | 0.783 | 2.301/2.299 |
| *MNIST-USPS* | Li et al. [2020] | 0.825 | 0.789 | 0.067 | 2.301/2.265 |
| | Reproduced | 0.835 | 0.785 | 0.018 | 2.301/1.579 |

Table 2: Reproduced and original quantitative results, for all metrics, on *Color Reverse MNIST* and *MNIST-USPS* dataset.

First of all, the reproduced accuracies on both datasets are very similar to the original values of Li et al. [2020]; differing 0.029 and 0.01 on *Color Reverse MNIST* and *MNIST-USPS* respectively. Secondly, similar to accuracy, the original and reproduced NMI values do not differ much; 0.88 on *Color Reverse MNIST* and 0.004 on *MNIST-USPS* . Thirdly, the reproduced balance on *Color Reverse MNIST* is close to the original; differing 0.02: however, the difference is larger on the *MNIST-USPS* dataset (0.049). Finally, the entropy values on *Color Reverse MNIST* are very similar in contrast to the original and reproduced entropy on *MNIST-USPS* .

### 4.2 Results beyond original paper

#### 4.2.1 Divergence Functions

Table 3 shows the results for different divergence functions as clustering regularizers.

| Dataset | Divergence Function | Accuracy | NMI | Balance | Entropy |
|---------|--------------------|----------|-----|---------|---------|
| *Color Reverse MNIST* | KL-divergence | 0.548 | 0.591 | 0.783 | 2.301/2.299 |
| | JS-divergence | 0.517 | 0.397 | 0.701 | 2.301/2.289 |
| | CS-divergence | 0.592 | 0.408 | 0.025 | 2.301/2.084 |
| *MNIST-USPS* | KL-divergence | 0.835 | 0.785 | 0.018 | 2.301/1.579 |
| | JS-divergence | 0.816 | 0.753 | 0.000 | 2.301/1.056 |
| | CS-divergence | 0.815 | 0.755 | 0.000 | 2.301/0.737 |

Table 3: Quantitative results for the *Color Reverse MNIST* and *MNIST-USPS* dataset, for all four metrics, with varying divergence measures.

On the *Color Reverse MNIST* dataset it can be observed that the accuracy do not differ significantly. Furthermore, using the CS-divergence seems to yield the highest accuracy. However, the NMI decreases significantly with JS- and CS-divergence as clustering regularizer. On top of that, the balance and entropy decrease significantly with CS-divergence. Using the JS-divergence also results in a decrease in balance and entropy on the *Color Reverse MNIST* dataset, even though that decrease is minor compared to the CS-divergence. In general, the KL-divergence outperforms the other two divergences on three of the four metrics on the *Color Reverse MNIST* dataset.

On the *MNIST-USPS* dataset, it can be seen that the difference in accuracy and NMI is even less significant compared to the *Color Reverse MNIST* dataset. However, on the *MNIST-USPS* dataset all four metrics decrease when using the JS- or CS-divergence instead of the KL-divergence. Moreover, the balance and entropy seem to decrease more significantly than the accuracy and NMI. In general, on the *MNIST-USPS* dataset the JS- and CS-divergence perform worse than the KL-divergence.

### 4.2.2 Corrupted Sensitive Attribute

The results of the corruption extension can be found in Table 4.

| Dataset | Corruption (in %) | Accuracy | NMI | Balance | Entropy |
|---|---|---|---|---|---|
| *MNIST* | 0.1 | 0.451 | 0.487 | 0.639 | 2.301/2.288 |
| | 0.4 | 0.342 | 0.314 | 0.001 | 0.837/2.258 |
| *Color Reverse MNIST* | 0.1 | 0.635 | 0.606 | 0.645 | 2.301/2.289 |
| | 0.4 | 0.474 | 0.483 | 0.002 | 2.164/2.198 |
| Both | 0.1 | 0.446 | 0.531 | 0.659 | 2.299/2.285 |
| | 0.4 | 0.313 | 0.213 | 0.000 | 1.615/1.583 |

Table 4: Quantitative results, on all four metrics, with varying corruption rates.

As can be seen, both the accuracy and the NMI decrease when data has been corrupted. However, the decrease in accuracy and NMI seems to be more significant when the *Color Reverse MNIST* dataset is corrupted. On top of that, the balance and entropy decrease as well when the data is corrupted. In general, a higher corruption leads to lower values on all metrics.

### 4.2.3 Pretrained Cluster Centers

The final extension researches the influence of the pretrained cluster centers on the utility and fairness of the clusters. The results for both datasets can be found in Table 5.

| Dataset | Pretrained | Accuracy | NMI | Balance | Entropy |
|---|---|---|---|---|---|
| *Color Reverse MNIST* | Yes | 0.548 | 0.591 | 0.783 | 2.301/2.299 |
| | No | 0.468 | 0.494 | 0.872 | 2.301/2.302 |
| *MNIST-USPS* | Yes | 0.835 | 0.785 | 0.018 | 2.301/1.579 |
| | No | 0.822 | 0.770 | 0.000 | 2.301/1.568 |

Table 5: Quantitative results for all metrics, on *Color Reverse MNIST* and *MNIST-USPS* datasets, with and without using pretrained cluster centers.

As can be seen, for both datasets accuracy and NMI are higher when pretrained cluster centers are used. The difference in accuracy is larger on the *MNIST-USPS* dataset, whereas the difference in NMI is smaller on this dataset, compared to *Color Reverse MNIST* . Moreover, the difference in balance on *MNIST-USPS* is not significant (0.018) while this is approximately five times larger (0.089) on the *Color Reverse MNIST* dataset. Finally, the entropy does not change significantly on both datasets.

## 5 Discussion

Our experimental results support the main claims of the original paper; namely that DFC is able to produce fair and clustering-favorable representations of large-scale and high dimensional data, such as images.

Furthermore, our extensions seem to add to the robustness of the model and strengthen the choices made by the original paper. First of all, the results of the different divergence functions show that both, CS- and JS-divergence, work but the default, KL-divergence, outperforms the two researched alternatives. Moreover, even though the *Color Reverse MNIST* dataset required the training of a new VAE and k-means clustering the results were still comparable; this speaks to the robustness of the algorithm that the original authors designed.

### 5.1 What was easy

The open source code of the authors was conveniently arranged. For example, the divergence function was put in the utils file, which made it easy to test other divergence functions as well. Also, the code had an implementation that randomly initialises cluster centers; to discard the pretrained cluster centers only modifications in the main file were needed. Once we understood the code base, the code structure became intuitive and easy to work with.

### 5.2 What was difficult

First of all, a difficulty while reproducing the research was caused by the coding style; due to the lack of comments it was difficult at the start to get a good understanding of the code. Secondly, we were required to download the data ourselves. However, these filenames and labels did not correspond to the included .txt-files by the authors. Therefore, the model did not learn and we were forced to produce our own `train_mnist.txt` and `train_usps.txt`. Thirdly, the algorithm uses pretrained models, a pretrained VAE, and a file with pretrained cluster centers. However, the authors solely provided these for one of the four datasets, namely *MNIST-USPS* . Thus, for *Color Reverse MNIST* we had to build our own VAE based on their structure and calculate our own cluster centers. The latter came with an extra difficulty since in the paper it is not stated how the clustering was performed. Therefore, we had to guess and chose k-means clustering. This made the reproduction of the *Color Reverse MNIST* dataset much harder than anticipated.

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
