# OpenReview forum: "Reproducibility Report: Deep Fair Clustering for Visual Learning"
_ML_Reproducibility_Challenge/2020 — RC2020_

### Official Review · AnonReviewer2 · 2021-02-28
**A careful and well organised report with extensions**

**Rating:** 9
**Confidence:** 4

**Review:**

This paper provides a holistic reproducibility report of the original paper on the algorithm named 'deep fair clustering' (DFC). The following issues are examined.

1. On the data sets 'Color Reverse MNIST' and 'MNIST-USPS' (they are the two data sets among four used in the original paper), DFC is tested against all the four metrics used in the original paper: 'Accuracy', 'NMI', 'Balance', and 'Entropy'. The mathematical details of these metrics are provided. In particular, a part of DFC and the data pre-processing are re-implemented since the corresponding code for the original paper is not accessible.

2. Extension by using no pre-trained cluster centres. It is shown that by discarding pre-trained cluster centres, DFC has a noticeable drop in performance.

3. Extension by using different divergence functions for regularisation. In particular, JS and CS divergences are used to replace the KL divergence used in the original paper. The comparison of performance is well documented.

4. Extension by using non-binary/corrupted sensitive attributes. It is shown that with corruption, DFC has a noticeable drop in performance in all the four metrics.
The paper is well organised, and the clear structure makes the report easy to follow.

Possible problems:

1. The words' fair' and 'effective' have both general English meaning and context-specific definitions, and both are tightly connected to machine learning and artificial intelligence. Using such words without giving a brief introduction leaves the readers confused. The confusion is not reduced, e.g., even after authors write 'feature representations are considered fair if they are statistically independent of sensitive attributes.'

2. The point of 'sensitive attributes' in section 2 is confusing. I am particularly lost at how this is related to turning the background of images to white or black.

3. Usually, the change of regularisers would significantly modify the performance of an algorithm. Examples include enhancing sparsity by l1 or l0 penalty, enhancing prediction accuracy by l2 penalty, and so on. I am not sure if it is fair (see, this word 'fair' has a different meaning than it has in the paper) to test the claims' robustness by changing regularisation.

4. How is the 'background' of an image or the 'background colour' defined?

5. A typo in line 149, p(x=q(x)) should be p(x)=q(x).


**Familiar With The Original Paper:**

I have not read the original paper

**Reproducibility Summary:**

Report has summary

---

### Official Review · AnonReviewer3 · 2021-03-01
**Deep Fair Clustering (DFC) for visual learning; Accurate and detailed report**

**Rating:** 8
**Confidence:** 4

**Review:**

The author(s) provide precise details in their report regarding the proposed algorithm and data set (MNIST) details. The objective function and model description are well defined. They provide every detail of the code and algorithms that have been used by other papers and provide the citation. The evaluation criteria; accuracy and NMI were used to evaluate cluster validity. They didn't discuss details of their result in Tables 3, 4, and 5, but they were understandable.

**Familiar With The Original Paper:**

I have not read the original paper

**Reproducibility Summary:**

Report has summary

---

### Decision · Program_Chairs · 2021-03-31

**Decision:**

Accept

**Comment:**

Selected for ReScience-C Journal Publication.